# EAPCR: A Universal Feature Extractor for Scientific Data without Explicit Feature Relation Patterns

## Abstract

Conventional methods, including Decision Tree (DT)-based methods, have been highly effective in scientific tasks, such as non-image medical diagnostics, system anomaly detection, and inorganic catalysis efficiency prediction. However, most deep-learning techniques have struggled to surpass or even match this level of success as traditional machine learning methods. The primary reason is that these applications involve multi-source, heterogeneous data, where features lack explicit relationships. This contrasts with image data, where pixels exhibit spatial relationships; textual data, where words have sequential dependencies; and graph data, where nodes are connected through established associations. The absence of explicit Feature Relation Patterns (FRPs) presents a significant challenge for deep learning techniques in scientific applications that are not image, text, and graph-based. In this paper, we introduce *EAPCR*, a universal feature extractor designed for data without explicit FRPs. Tested across various scientific tasks, EAPCR consistently outperforms traditional methods and bridges the gap where deep learning models fall short. To further demonstrate its robustness, we synthesize a dataset without explicit FRPs. While Kolmogorov–Arnold Network (KAN) and feature extractors like Convolutional Neural Networks (CNNs), Graph Convolutional Networks (GCNs), and Transformers struggle, EAPCR excels, demonstrating its robustness and superior performance in scientific tasks without FRPs.

## 1 Introduction

In various scientific applications, such as non-image medical diagnostics, system anomaly detection, inorganic catalysis efficiency prediction, and etc., traditional machine learning techniques, such as Decision Tree (DT) (Ali et al., 2012) and DT-based method (e.g., Random Forest (RF) and Extreme Gradient Boosting (XGBoost)), have been reported as highly effective (Coşkun & Kuncan, 2022; Mutlu et al., 2023; Alizargar et al., 2023; Schossler et al., 2023; Mallioris et al., 2024). In contrast, few studies have reported deep learning models as the best-performing methods, indicating that more complex deep learning techniques, such as Convolutional Neural Network (CNN)s (LeCun et al., 1998), Graph Convolutional Network (GCN)s (Kipf & Welling, 2016; Bronstein et al., 2017), and Transformers (Vaswani, 2017), have not shown the same level of success in those scientific applications.

The primary reason deep learning techniques often underperform in scientific applications is that the data in these fields differ significantly from traditional tasks like images, text, and graphs. For example, in non-image medical diagnostics, patient data come from diverse sources, such as physical measurements (e.g., weight, blood pressure) and chemical tests (e.g., glucose levels) (Fig. 1-a-1). Unlike pixels in images or words in text, which have spatial or sequential relationships (LeCun et al., 1998; Bronstein et al., 2017) (Fig. 1-b-1), or nodes in graphs with known connections (Kipf & Welling, 2016; Bronstein et al., 2017) (Fig. 1-b-2), features in scientific data lack such explicit relationships. This absence of explicit feature relationships is common across various scientific tasks, such as system anomaly detection (Wang et al., 2023; Tian, 2023) (Fig. 1-a-2) and inorganic catalysis efficiency prediction (Sun et al., 2024) (Fig. 1-a-3), where features are collected from heterogeneous sources, such as electrical signals and temperature, and have different units, like pH levels and illumination time.

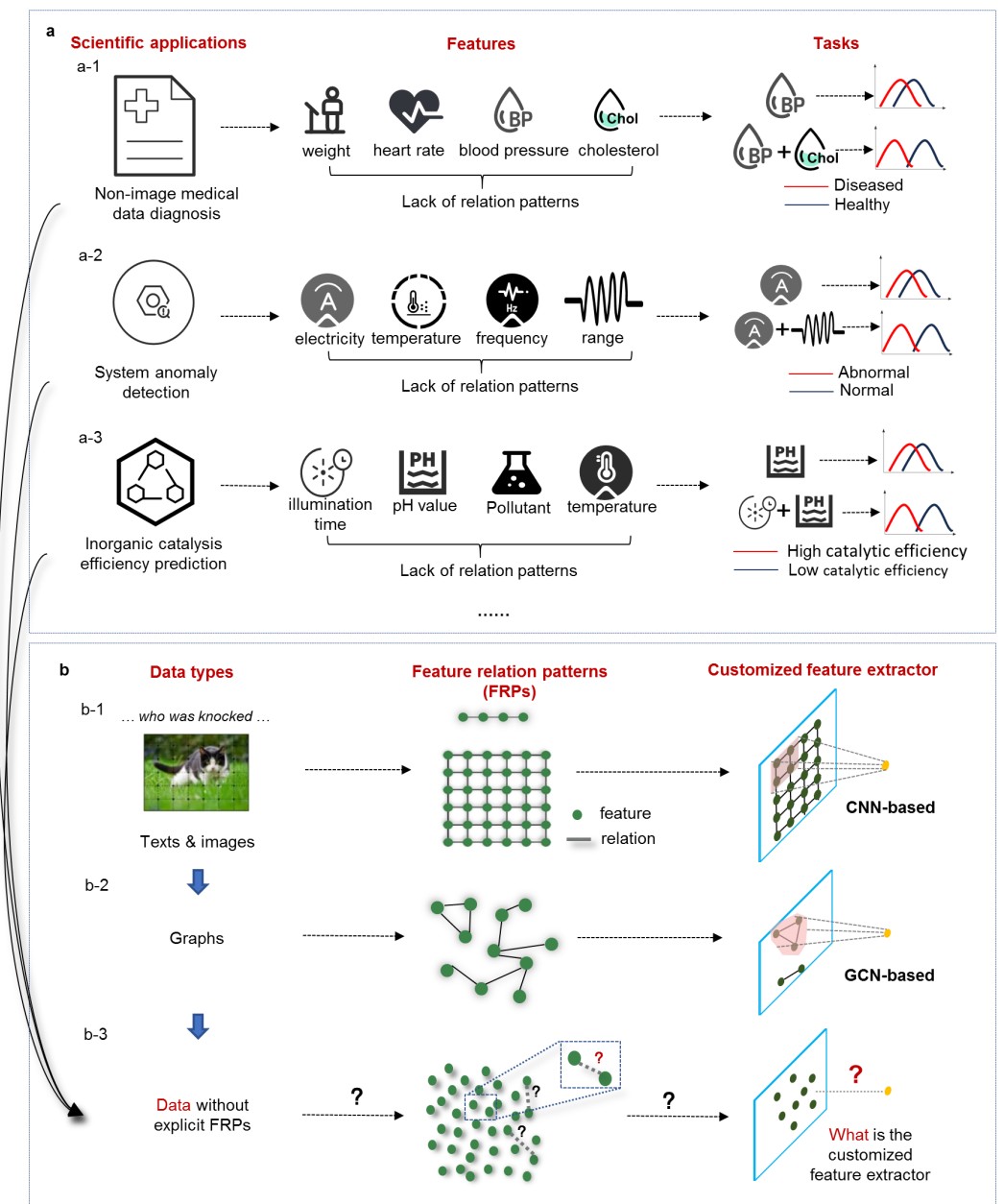

Figure 1: Motivation for designing feature extractors for data without FRPs.

The feature relationship patterns, including spatial relationships and connections, are referred to as **F**eature **R**elation **P**atterns (FRPs) in this work. FRPs are critical for deep learning-based feature extractors as they contain essential information about feature associations or correlations. For instance, in image and text data, spatial relationships between pixels and words reveal correlations where nearby features have stronger associations and distant ones weaker. This inherent relational information enables feature extractors to quickly identify important combinations of strongly interacting features. For example, CNNs in image data leverage spatial relationships to focus on local feature patterns while ignoring irrelevant non-local ones (Bronstein et al., 2017; Yun et al., 2023) (Fig. 1-b-1). Similarly, GCNs use adjacency matrices in graph data to capture meaningful node connections (Bronstein et al., 2017; Schlichtkrull et al., 2018) (Fig. 1-b-2). However, when explicit FRPs, like spatial relationships or known connections, are absent, deep learning methods often strug-

gle (Fig. 1-b-3) as assumed FRPs may not match the actual implicit relationships. This raises a key question:

*How can we design universal feature extraction modules for data that lack explicit FRPs?*

In this paper, we propose a feature extraction module, *EAPCR*, designed as a universal feature extractor for data without explicit FRPs. Traditional feature extraction modules rely on known FRPs to distinguish between important and unimportant feature combinations. However, without the guidance of FRPs, EAPCR adopts a different approach. First, it exposes all possible FRPs. Second, it accelerates the sampling of these combinations to ensure a wide range of feature interactions are evaluated, allowing it to effectively identify important combinations of strongly interacting features.

To evaluate EAPCR's effectiveness, we apply it to various scientific domains, including non-image medical diagnostics (Anderies et al., 2022), system anomaly detection (Tian, 2023), and inorganic catalysis efficiency prediction (Liu et al., 2022). EAPCR consistently outperforms traditional methods in such tasks lacking explicit FRPs. To further assess its robustness, we synthesize a dataset without explicit FRPs, where models like CNNs (LeCun et al., 1998), GCNs (Kipf & Welling, 2016; Bronstein et al., 2017), Transformers (Vaswani, 2017), and KAN (Liu et al., 2024b;a) struggle, while EAPCR excels in capturing meaningful features.

In summary:

- EAPCR is designed as a universal feature extractor for tasks lacking explicit FRPs, addressing a critical gap in deep learning for scientific applications. It consistently outperforms other models across various real-world scientific applications.

- We synthesize a representative dataset to investigate the challenges of modeling without FRPs, revealing the limitations of traditional methods and validating the robustness and effectiveness of EAPCR.

**Related Works** *Feature engineering at the early stage:* Feature engineering (Bengio et al., 2013) plays a crucial role in improving the accuracy of classification models. Early approaches primarily focus on addressing feature redundancies and nonlinear relationships. For instance, principal component analysis (PCA) (Abdi & Williams, 2010) reduces linear correlations, while nonlinear methods like nonlinear PCA (Linting et al., 2007) and autoencoders (Bank et al., 2023) handle redundancies through nonlinear transformations. Although classifiers such as Support Vector Machine (SVM)s (Hearst et al., 1998), Multi-layer Perceptron (MLP)s (Rumelhart et al., 1986), and more recent models like Kolmogorov–Arnold Network (KAN) (Liu et al., 2024b;a) can manage complex nonlinear feature relationships, their performance heavily depends on how input data is represented. For example, using large pre-trained models to encode images improves classification and retrieval accuracy by emphasizing critical features like edges and shapes (Liu et al., 2023; Holliday & Dudek, 2020; Zhou et al., 2024). Therefore, more advanced feature extraction techniques are required to go beyond capturing nonlinear relationships, further refining feature representations for better performance.

*Feature engineering for data with FRPs:* Accurately capturing implicit correlations between features is essential for effective classification. For example, determining obesity cannot solely rely on weight; height must also be considered to provide a more accurate assessment. In more complex scenarios, classification depends on interactions between features, where their joint contribution exceeds the sum of their individual effects (Koh & Liang, 2017; Ali et al., 2012; Beraha et al., 2019; Deng et al., 2022). This is why traditional classifiers, like MLPs (Rumelhart et al., 1986), rely on advanced feature extractors to improve performance by identifying complex feature interactions. In this vein, recent advancements, such as ConvNeXt (Woo et al., 2023), Bidirectional Encoder Representation from Transformers (BERT) (Devlin, 2018), Generative Pre-trained Transformer (GPT) (Radford et al., 2019), Vision Transformer (ViT) (Dosovitskiy, 2020), and Temporal Fusion Transformers (TFT) (Lim et al., 2021), efficiently capture interaction patterns of features in structured or Euclidean data like images and texts. For non-Euclidean data, techniques like manifold learning (McInnes et al., 2018; Tenenbaum et al., 2000) and Graph Neural Network (GNN)s, including Graph SAmple and aggreGatE (GraphSAGE) (Hamilton et al., 2017) and Deep Graph Convolutional Neural Network (DGCNN) (Wang et al., 2019), address unique challenges. Regardless of the data type, these methods rely on explicit FRPs (e.g., spatial, sequential, or relational

connections) of data, which contain the implicit feature correlations essential for effective feature extraction.

*Feature engineering for data without FRPs:* As discussed earlier, many scientific tasks lack explicit FRPs. Machine learning techniques like Decision Tree (DT) and DT-based methods perform well in these tasks by handling various data types (numerical and categorical) and automatically capturing interaction effects between features, as each decision split evaluates the relationship between a feature and the target variable (Gregorutti et al., 2017). In contrast, deep learning models, such as CNNs, GCNs, and Transformers, struggle due to their reliance on predefined FRPs. For example, in heart failure and maternal health risk prediction, the best-performing models are RF (Coşkun & Kuncan, 2022) and DT (Mutlu et al., 2023). Similarly, in hepatitis C (Alizargar et al., 2023), $TiO_2$ photocatalytic degradation (Schossler et al., 2023), and centrifugal pump health status prediction (Mallioris et al., 2024), XGBoost, another DT-based method, consistently outperforms deep learning methods. Despite advancements in multimodal techniques (Lahat et al., 2015) and methods for non-Euclidean data (Bronstein et al., 2017), feature heterogeneity does not always align with distinct modalities. Even features from the same source may lack explicit FRPs, rendering multimodal approaches ineffective. Moreover, inconsistent feature dimensions complicate the definition of feature distances in Euclidean space. For instance, determining how a 1 kg increase in weight correlates with a change in height for a particular disease is non-trivial. Data without explicit FRPs differ from traditional Euclidean and non-Euclidean data (Bronstein et al., 2017), making deep learning techniques less effective in these applications.

## 2 EAPCR: A FEATURE EXTRACTOR WITHOUT THE NEED OF EXPLICIT FEATURE RELATION PATTERNS

In many applications, identifying feature combinations with strong interactions is not difficult, as feature extraction modules like CNNs and GCNs use predefined FRPs to narrow the range of possible combinations. For example, CNNs leverage spatial relationships between pixels to efficiently sample local regions and filter out weak feature interactions, quickly identifying critical patterns like textures in image recognition. However, when explicit FRPs are absent, searching for important feature interactions becomes more random and inefficient. Traditional methods often fail in these cases because the FRPs chosen by feature extraction modules may not align with the implicit patterns in the data. Additionally, the sample complexity increases exponentially with the number of features, making exhaustive search impractical. Unlike these traditional approaches, EAPCR exposes all feature combinations to ensure no fundamental interaction patterns of features are missed (Fig. 2-b), then optimizes the efficiency of combination sampling to address this challenge (Fig. 2-c).

Further discussion about why FRPs is important gives in App. F, where the relationship between feature interaction, feature correlation, and FRPs is discussed.

### 2.1 EXPOSE POSSIBLE FEATURE RELATIONS: EMBEDDING AND BILINEAR ATTENTION

Unlike existing works that use one-hot encoding for categorical feature representation (Seger, 2018) and bilinear attention to focus on interactions between two input modalities (Fukui et al., 2016), we leverage **E**mbedding (Mikolov, 2013) and bilinear **A**ttention (Kim et al., 2018) to construct a correlation matrix.

For an input with $N$ features, we first convert each feature into a categorical (string-based) one. Categorical features, like gender or catalyst substrate, remain unchanged, while numerical features are discretized into categories (e.g., "high", "medium", "low") based on context-specific thresholds. These thresholds are chosen to balance between overly fine granularity, which leads to sparse categories, and overly coarse granularity, which reduces feature separation. For example, temperature might be categorized into "very high", "high", "medium", "low", and "very low". This process generates an input matrix of shape $[N, 1]$, where each element is an integer index, assigned via a dictionary mapping categorical values to indices.

The embedding operation substitutes each component by a corresponding dense vector, giving $E$ with shape $[N, E_s]$, where $E_s$ is the embedding size. Then, we consider the bilinear attention defined as:

$$A = \text{Tanh}(EE^\top), \tag{1}$$

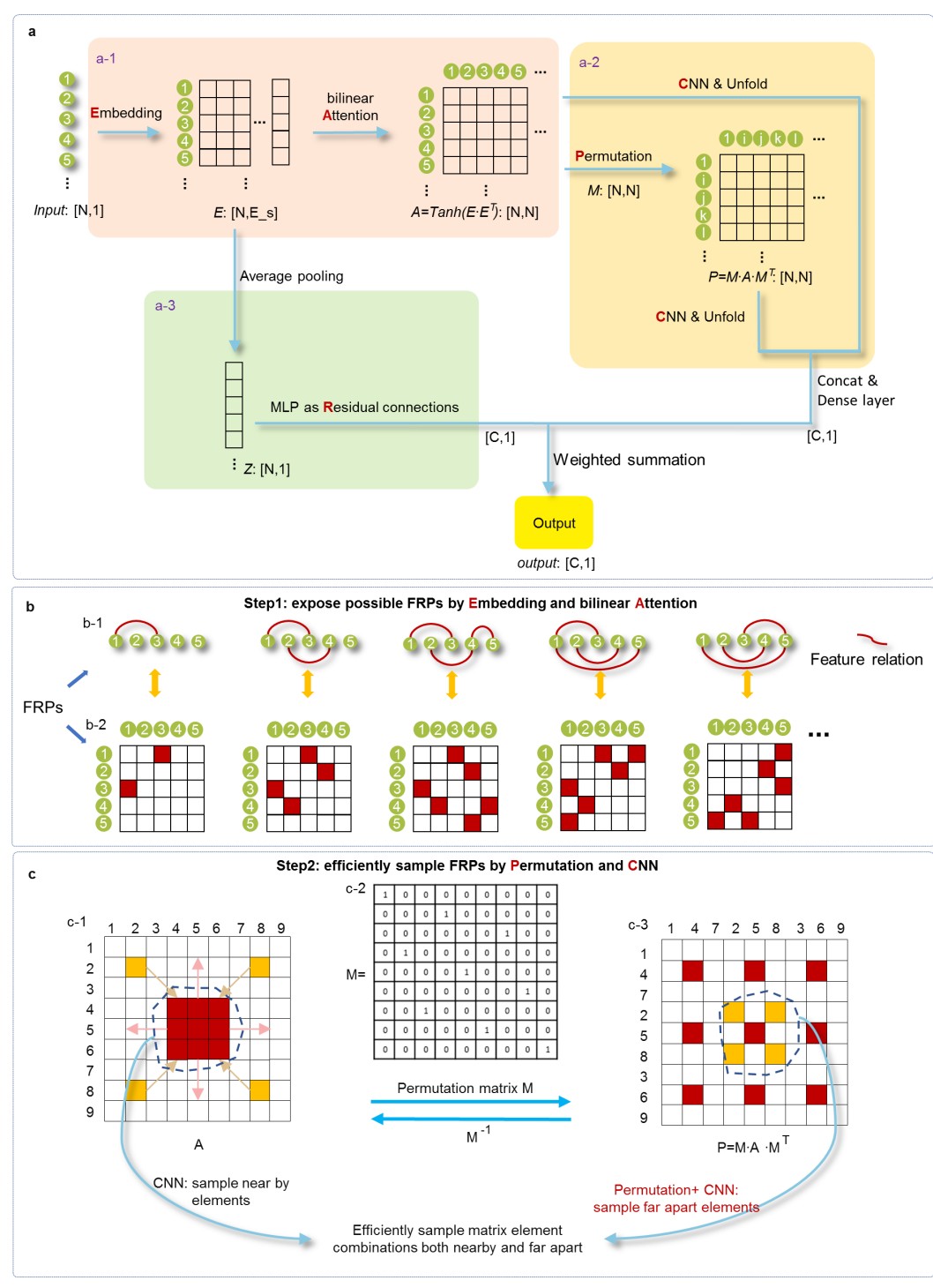

Figure 2: The illustration of the method. (a) an overview of EAPCR. (b) an illustration of how Embedding and bilinear Attention can expose all possible FRPs. (c) an illustration of how the permuted CNNs considers combinations of originally close matrix elements as well as combinations of originally distant elements.

where the matrix $A$ with shape $[N, N]$ is the constructed correlation matrix (Fig. 2-a-1). Tanh($x$) is Hyperbolic Tangent function. The matrix $A$ is important, because each element in the matrix

represents the relationship between two features, with any combination of these elements exposing corresponding potential FRPs. Thus, by using matrix $A$, all possible relation patterns between features are being exposed in a matrix form, as shown in Fig. 2-b. The target combinations that consist of features with strong interactions are encapsulated within the matrix $A$.

Additionally, the embedding process not only captures feature relationships but also helps the model understand nuances across multi-source heterogeneous features.

## 2.2 EFFICIENTLY SAMPLE FEATURE RELATIONS: PERMUTED CNN

To identify the target combinations that consist of features with strong interactions within the matrix $A$, we leverage CNN to sample such combinations of elements in matrix $A$ defined in equation 1. Because CNNs can efficiently focus on local elements within a matrix (Bronstein et al., 2017). Instead of expanding the receptive field of CNNs by increasing kernel size or layers, we propose a **P**ermuted **C**NN that efficiently samples diverse element combinations from the matrix $A$.

*The designed permutation matrix $M$.* The designed permutation $M$ rearranges the matrix elements of $A$, bringing originally distant elements closer and pushing originally close elements further apart, as shown in Fig. 2-c. The details of constructing the designed permutation matrix $M$ can be found in App. A. We apply a designed permutation matrix $M$ on $A$ giving a new matrix $P$, defined as:

$$P \triangleq MAM^\top. \tag{2}$$

*The permuted CNN.* The permuted CNN architecture is designed to capture both local and non-local relationships within the matrix elements by applying the CNN to two different representations of the matrix $A$, as illustrated in Fig. 2-a-2. Specifically, the CNN is applied to the raw matrix $A$ as well as to a permuted version of $A$, that is $P$ in equation 2. The CNN varies on different tasks but is only equipped with a lightweight structure (e.g., a two-layered architecture with kernel sizes of $3 \times 3$ and channel numbers of 8 and 16). Afterward, the outputs from the CNN operations on both the raw matrix $A$ and the permuted matrix $P$ are transformed into vectors. These vectors are then concatenated, resulting in a single feature vector of size $[C, 1]$ by dense connection, where $C$ is the number of classes.

Moreover, by incorporating existing feature extraction modules, such as MLP, as a **R**esidual connection, we combine the strengths of different feature extractors to further enhance feature extraction efficiency. As shown in Fig. 2-a-3, the average pooling is applied on $E$, resulting in a vector $z$ with shape $[N, 1]$. Then, an MLP transforms $z$ into a vector with shape $[C, 1]$. Therefore, the residual networks also help the model better train the embedding vectors.

## 3 MAIN EXPERIMENT RESULTS

In Sec. 3.1, we present experimental results on real-world applications, including *non-image medical diagnostics*, *inorganic catalysis efficiency prediction*, and *system anomaly detection*. The EAPCR can also be used in a wider range of scientific applications beyond experiments mentioned in this section; see Tab. 11, Tab. 12, Tab. 13, Tab. 14, and Tab. 15 in App. E for more discussion. In summary, the experiment results show that EAPCR as a deep learning method outperforms traditional deep/non-deep methods, including the SOTA DT-based methods, demonstrating the effectiveness of the proposed EAPCR method for tasks involving data without explicit FRPs.

In Sec. 3.2, to investigate the challenges of modeling without FRPs, we construct an illustrative dataset lacking explicit FRPs. The results reveal the limitations of traditional deep learning methods, such as CNN (LeCun et al., 1998), GCN (Kipf & Welling, 2016; Bronstein et al., 2017), Transformer (Vaswani, 2017), MLP (Rumelhart et al., 1986), and KAN (Liu et al., 2024a), which are often underutilized in scientific tasks. We then validate the robustness and effectiveness of EAPCR. This dataset highlights the shortcomings of traditional models in tasks that lack explicit FRPs while demonstrating EAPCR's superior performance in such scenarios.

### 3.1 BENCHMARKING COMPARISON OF EAPCR ON SCIENTIFIC TASKS

**Non-image medical diagnostics:** In this study, we validated the proposed EAPCR method using the UCI Cleveland heart disease dataset (Janosi et al., 1989). The dataset consists of 13 feature

attributes (including Sex, Age, and CP [Chest Pain] Type) and one categorical attribute, comprising eight categorical (string-based) and five numerical features (details in App. E). Following the data processing approach outlined in Sec. 2.1, all features were converted to categorical form, and a dictionary was created to map each categorical feature to an integer index. These indices were then mapped to 128 dimensions using an embedding mechanism and fed into the model for training. We benchmarked the EAPCR method against various machine learning algorithms, including SVM, Naive Bayes algorithm (NB), logistic regression, DT, and $k$-Nearest Neighbor algorithm (KNN), as presented in Tab. 1, based on the study by Anderies et al. (Anderies et al., 2022). The results show that EAPCR outperforms conventional machine learning techniques.

Table 1: Comparison of our method with others in the diagnosis of non-image medical data.

| Method | Accuracy | Precision | Recall | F1 Score |
|---|---|---|---|---|
| DT | 70% | 86% | 63% | 72% |
| KNN | 78% | 90% | 74% | 81% |
| Logistic Regression | 83% | 94% | 79% | 86% |
| NB | 83% | 96% | 76% | 85% |
| SVM | 85% | 97% | 79% | 87% |
| **EAPCR** | **93%** | **97%** | **92%** | **94%** |

**Inorganic catalysis efficiency prediction:** In this study, we used the $TiO_2$ photocatalysts dataset from Liu et al. (2022) to evaluate the performance of our EAPCR model in inorganic catalysis applications. The dataset contains nine attributes, including dopant, molar ratio, calcination temperature, and pollutant, with a total of 760 samples (details in App. E). Using a 256-dimensional embedding, we benchmarked EAPCR against LightGBM, the best-performing model in Liu et al. (2022). As shown in Tab. 2, EAPCR outperforms LightGBM in terms of the $R^2$ metric.

Table 2: Comparison of our method with others in the Inorganic catalysis data set.

| Model | MAE | MSE | RMSE | $R^2$ |
|---|---|---|---|---|
| Linear Regression | $0.513 \pm 0.104$ | $0.601 \pm 0.237$ | $0.762 \pm 0.401$ | $0.048 \pm 0.014$ |
| RF | $0.235 \pm 0.062$ | $0.180 \pm 0.126$ | $0.417 \pm 0.148$ | $0.805 \pm 0.035$ |
| XGBoost | $0.145 \pm 0.103$ | $0.086 \pm 0.034$ | $0.293 \pm 0.136$ | $0.884 \pm 0.024$ |
| LightGBM (Liu et al., 2022) | / | / | / | 0.928 |
| **EAPCR** | **$0.128 \pm 0.003$** | **$0.041 \pm 0.001$** | **$0.203 \pm 0.004$** | **$0.937 \pm 0.003$** |

**System anomaly detection:** In our study on system anomaly detection, we used sensor data from the Kaggle dataset (https://www.kaggle.com/datasets/umerrtx/machine-failure-prediction-using-sensor-data), which is designed for predicting machine failures in advance. The dataset consists of 944 samples with nine feature attributes, including sensor readings like footfall and temp-Mode, and a binary target attribute (1 for failure, 0 for no failure) (details in App. E). Using a 64-dimensional embedding, we benchmarked our model against RF, Logistic Regression, SVM, and Gradient Boosting. As shown in Tab. 3, our model outperformed the others across all metrics.

## 3.2 BENCHMARKING COMPARISON OF EAPCR ON SYNTHESIZED DATA

**Synthesize data without explicit FRPs.** In image recognition, the correlation between pixels and their spatial positions is typically consistent, with nearby pixels having higher correlations and distant ones having lower correlations (see App. B). Previous research (Yun et al., 2023) shows that random transformations can disrupt this spatial relationship, such as by shuffling pixel positions. However, our approach differs by using a carefully designed permutation, as outlined in Sec. 2.2, which strategically moves adjacent matrix elements further apart and brings distant elements closer

Table 3: Comparison of our method with others in the Sensor data-based system anomaly detection tasks

| Algorithm | Accuracy | Precision | Recall | F1-score |
|---|---|---|---|---|
| RF | 87.83% | / | / | / |
| Logistic Regression | 87.83% | / | / | / |
| SVM | 87.83% | / | / | / |
| Gradient Boosting | 88.89% | 87.50% | 88.51% | 88.00% |
| **EAPCR** | **89.42%** | **87.64%** | **89.66%** | **88.64%** |

(Fig. 3-a). As an example, we used the handwritten digit recognition MNIST dataset (LeCun et al., 1998) to generate data without explicit FRPs, see Figs. 3-b and 3-c.

This designed permutation more effectively disrupts spatial relationships while visually maintaining them, thereby concealing underlying FRPs more thoroughly compared to random permutations (Yun et al., 2023). The proof can be found in Fig. 3-b, where CNN performance degrades progressively, demonstrating that the designed permutation more effectively breaks spatial correlations. This is why we applied the designed permutation when synthesizing the dataset.

Although both the permuted CNN in EAPCR and the synthesized data use a permutation matrix that disrupts the original spatial relationships, they are not identical. For instance, the synthesized data matrix is $[28, 28]$, while the model's matrix is $[784, 784]$, ensuring the model has no prior knowledge of hidden FRPs and preventing leakage. Further evidence comes in Fig. 3-b, where EAPCR performs consistently across raw data, randomly permuted data, and data with designed permutation.

We conducted extensive experiments on both the original data and the synthesized data, using existing machine learning models, mainstream deep learning models, and our proposed EAPCR model. For convenience, we refer to these two datasets as the "raw image (data with predefined FRPs)" and "synthesized data (data without explicit FRPs)," since the format of the synthesized data is no longer important.

**Performance of different models on data with FRPs and synthesized data without explicit FRPs.** We summarize the results of this section in Fig. 3-c, where the horizontal axis represents the parameters of different models and the vertical axis shows the accuracy of these models on Data With/Without explicit FRPs. Fig. 3-c is essential to understanding the limitations of traditional methods and the robustness and effectiveness of EAPCR.

*MLP as the baseline.* The MLP is a basic neural network architecture that uses fully connected layers to process input data. As shown in Fig. 3-c-1, MLP performs similarly on tasks with and without explicit FRPs. Thus, we use MLP's performance as a baseline to assess the feature extraction effectiveness of other models. Models that exceed MLP's performance demonstrate effective feature extraction capabilities, while those with similar or lower performance indicate insufficient feature extraction.

*EAPCR demonstrated superior performance.* Interestingly, on raw data, both the CNN (CNN+MLP) and the ensemble model CNN+KAN exhibited strong performance, as shown in Fig. 3-c-3 and Fig. 3-c-8, respectively. However, when evaluated on synthesized data, only EAPCR and EACR (the ablation model of EAPCR where permuted CNN is eliminated) obviously outperforms the MLP, with results depicted in Fig. 3-c-7 for EAPCR and Fig. 3-c-6 for EACR. Notably, EAPCR achieved $94.5\%$ accuracy with 37,355 parameters, outperforming its ablation model EACR, which achieved $94.3\%$ accuracy with 59,361 parameters. Other models, such as the RF, CNN, and GCN (uses the data's correlation matrix as its adjacency matrix), only slightly surpassed the MLP. Their performances are shown in Fig. 3-c-2 for CNN, Fig. 3-c-4 for GCN, and Fig. 3-c-5 for RF. It is important to note that the GCN's performance declined significantly when provided with a randomly generated adjacency matrix instead of the correlation matrix (Fig. 3-c-4). Transformer (Vaswani, 2017) (Fig. 3-c-11) and KAN (Liu et al., 2024a) (Figs. 3-c-10 and 3-c-9) do not exhibit obvious impact on synthesized data without FRPs. More details of the models' parameter settings are given in the Tabs. 5 and 6 and in App. D.

*EAPCR recovers hidden FRPs.* The reason EAPCR demonstrates supreme performance compared to other methods is that it successfully reconstructs the implicit FRPs in the synthesized dataset. We verify this by comparing the correlation matrix recovered by EAPCR with the original correlation matrix. The details about the result are shown in App. G.

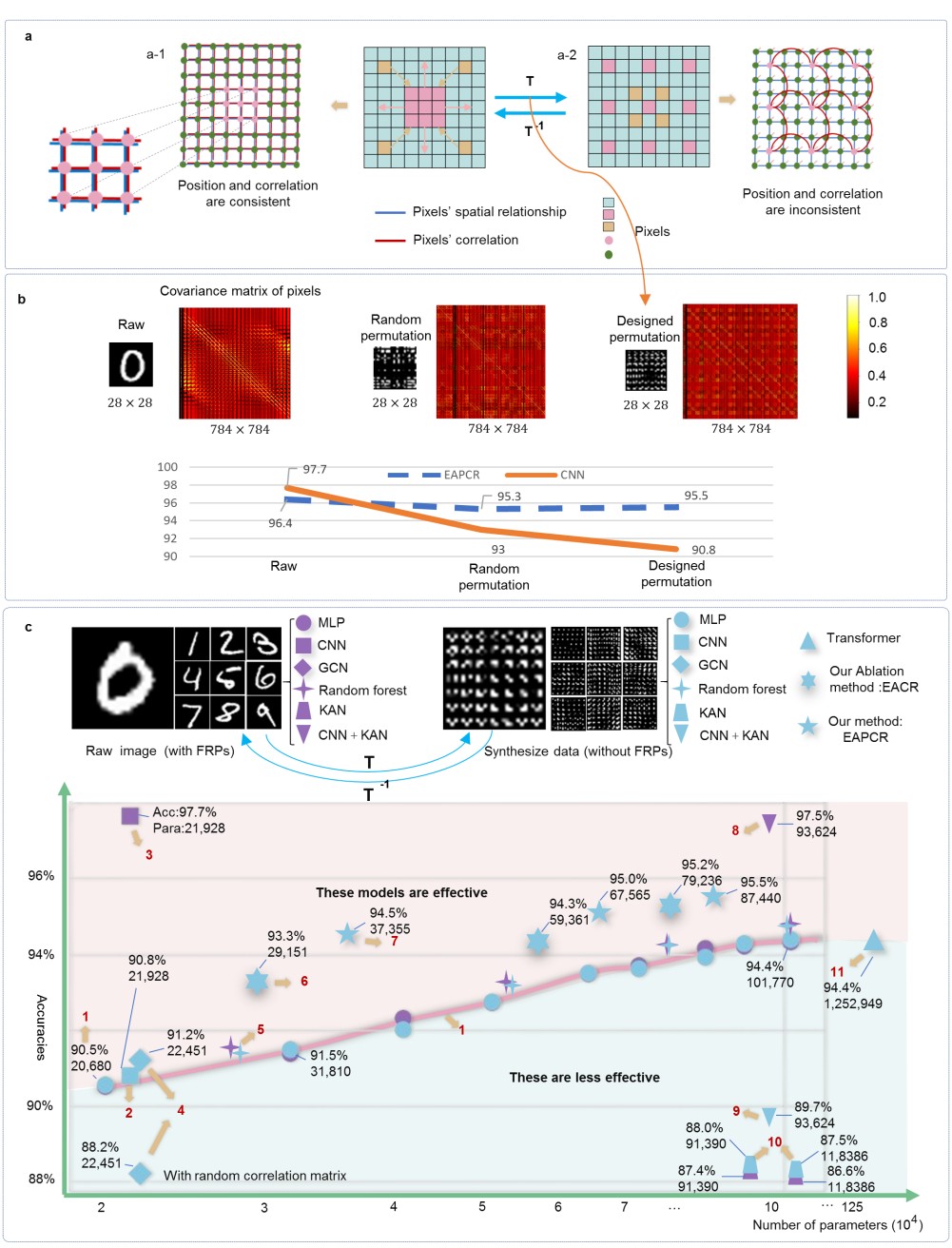

Figure 3: Results on synthesized data without explicit FRPs. (a) Illustration of synthesized data: In the raw image, pixel correlations align with spatial positions (a-1), but in the synthesized data, the spatial correlations breaks (a-2). (b) Experimental results confirm that EAPCR's strong performance is not due to the use of a similarly designed permutation when synthesizing the data. (c) Comparison of various methods on data with FRPs and synthesized data without explicit FRPs.

### 3.3 ROBUSTNESS AND ABLATION STUDY

To further validate the robustness of EAPCR, we synthesized additional datasets beyond the MNIST dataset to Flower (Nilsback & Zisserman, 2008), ImageNet (Deng et al., 2009), and CIFAR-10 (Krizhevsky & Hinton, 2009) datasets. We also conducted experiments comparing EAPCR to EACR, an ablation model without permutation but with larger convolutional kernels and more layers, to demonstrate that EAPCR's permuted CNN is more effective than simply increasing the capacity of standard CNNs. In addition, experiments in App. C show that the permutation we designed in the permuted CNN outperforms a random one. The results (Fig. 4) show: 1) traditional CNN-based algorithms, such as ConvNeXt-V2 (Woo et al., 2023), perform well on data with FRPs but fail on data without explicit FRPs; 2) EAPCR consistently performs well across all three synthesized datasets without explicit FRPs; and 3) EAPCR outperforms EACR, even though EACR uses larger kernels and more layers.

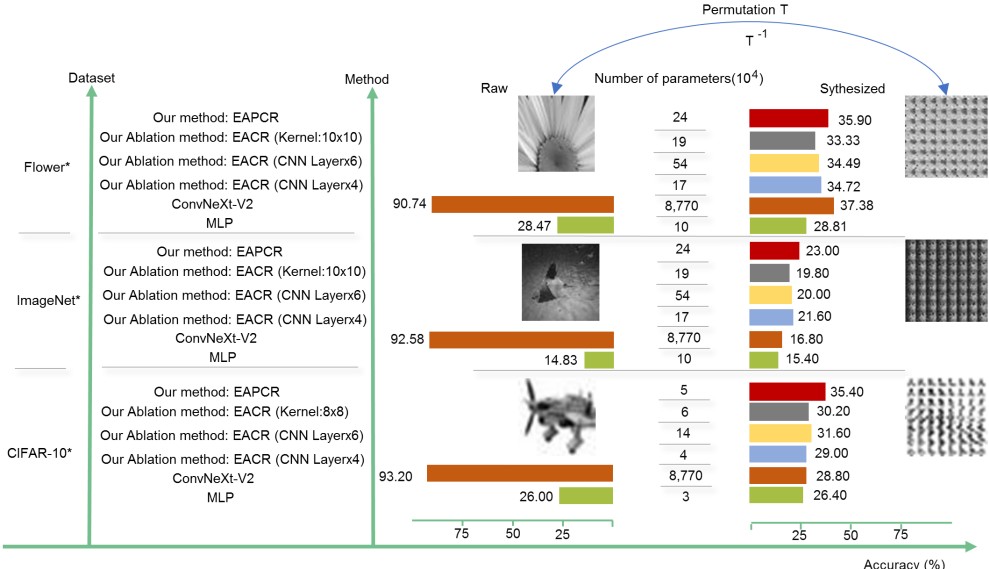

Figure 4: The comparison and ablation experiments conducted on the more synthesized data. Asterisks indicate that only subsets of gray data were used for experimental efficiency. This involves employing limited random sampling to create combined training and testing datasets. For example, from ImageNet, 10 categories, each comprising 400 randomly selected images, were selected for training and 50 for testing, optimizing both time and computational resources.

## 4 CONCLUSION

The absence of explicit Feature Relation Patterns (FRPs) presents a significant challenge in many scientific tasks that are often overlooked by the ML community. This limitation contributes to the underperformance of deep learning methods compared to traditional methods, such as Decision Tree-based (DT-based) machine learning approaches, in scientific applications. To address this issue, we introduce an innovative method, EAPCR, specifically designed for data lacking FRPs. We evaluate the effectiveness and efficiency of EAPCR across a variety of real-world scientific tasks and demonstrate that it consistently outperforms established methods on several scientific datasets. Additionally, we synthesize a dataset that deliberately excludes explicit FRPs to further assess the performance of EAPCR. The results demonstrate that EAPCR outperforms CNN, GCN, MLP, RF, Transformer, and KAN on the dataset without explicit FRPs. Our findings underscore the potential of EAPCR as a robust solution for scientific tasks lacking explicit FRPs, bridging the gap where deep learning models fall short and paving the way for enhanced data analysis in this domain.

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

## A    GENERATING PERMUTATION MATRIX M

Constructing the designed permutation matrix in Sec. 2.2 is a key aspect of this paper. Although the method is relatively simple, we describe the process in detail in the appendix. The core idea is to create a reversible operation that rearranges the positions of $N$ elements $(1, 2, 3, \cdots, N)$, ensuring that originally adjacent elements are no longer adjacent while non-adjacent elements become adjacent. This generates the reversible permutation matrix $M$.

The process works as follows: First, arrange the $N$ elements in order into an $R \times L$ matrix, where $N = R \times L$ with $R$ and $L$ being roughly equal in size. Transpose of this matrix and then reshape it into an $N \times 1$ vector. This new sequence represents the transformed positions of the original data. Next, create an all-zero matrix $M$ of size $N \times N$. Using the transformed positions, place 1s in the corresponding row and column positions of $M$, resulting in the reversible permutation matrix.

For example, with $N = 9$, arrange the numbers 1 to 9 into a $3 \times 3$ matrix $[[1, 2, 3], [4, 5, 6], [7, 8, 9]]$. Transposing the matrix and reshaping it gives the new sequence $[1, 4, 7, 2, 5, 8, 3, 6, 9]$. Using this sequence, place 1s in the appropriate positions in a $9 \times 9$ matrix $M$. The result is a permutation matrix where the distance between two adjacent elements in the new sequence is at least 3, meaning originally adjacent elements are no longer adjacent, and previously distant elements are now adjacent. Further details can be found in the code.

## B    ANALYSIS OF PIXEL DISTANCE VS. PIXEL RELATIONS

To investigate the relationship between pixel distance and correlation in images, we randomly selected 200 images from 10 subfolders of the MNIST handwriting dataset. We calculated the Pearson correlation coefficient and mutual information for different pixel pairs based on their distance. Using the reference point $(5, 5)$, we compared its relationship with all other pixels in each image. The results were sorted by distance, and for each unique distance, we retained only the highest correlation coefficient. We then plotted the variation of the Pearson correlation coefficient and mutual information with pixel distance at distances of $1$, $\sqrt{2}$, $2$, etc., as shown in Fig. 5.

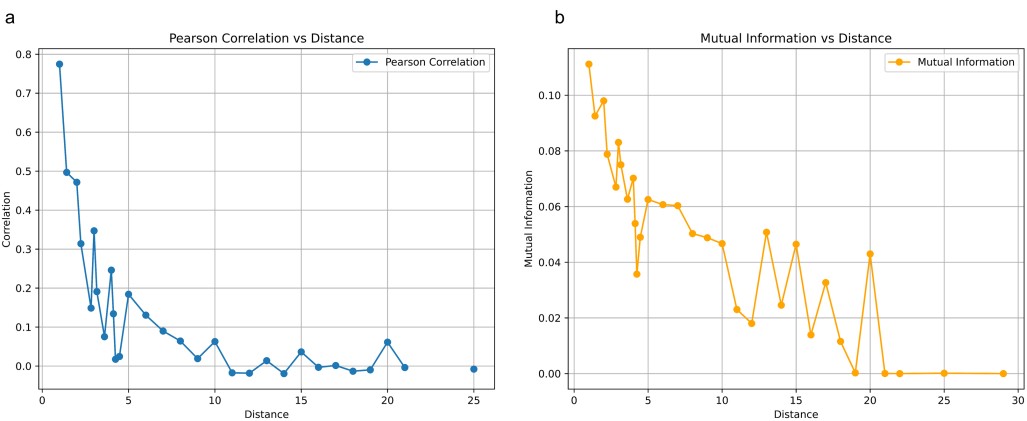

Figure 5: Analysis of inter-pixel distance and statistical correlations. (a) Relationship between the Pearson correlation coefficient and inter-pixel distance. (b) Relationship between mutual information and inter-pixel distance.

## C    THE PERMUTED CNN WITH DESIGNED PERMUTATION OUTPERFORMS THAT WITH RANDOM PERMUTATION

To demonstrate that the designed permutation used in our permuted CNN (see Sec. 2.2) outperforms random permutation, we conducted comparative experiments on synthesized data based on

the MNIST dataset. The experimental results are presented in Tab. 4. The results show that the designed permutation performs better, as it maximally separates nearby matrix elements while bringing distant ones closer, enabling the CNN to effectively sample both nearby and distant matrix elements.

Table 4: Experiment of permuted CNN with designed permutation versus that with random permutation.

| Method | Parameters | Accuracies |
|---|---|---|
| EAPCR with designed permuted CNN | 37355 | **94.5%** |
| EAPCR with random permuted CNN | 37355 | 93.2% |

## D    EXPERIMENT DETAILS: SYNTHESIZED DATASET WITHOUT EXPLICIT FRPS

An illustration of the EAPCR for the synthesized data of the handwritten digital MNIST dataset is given in Fig. 6. Tab. 5 shows the details of the structure and parameters of our EACPR and ablation model EACR. Tab. 6 shows the details of the structures and parameters of other models used in the experiment of synthesized data.

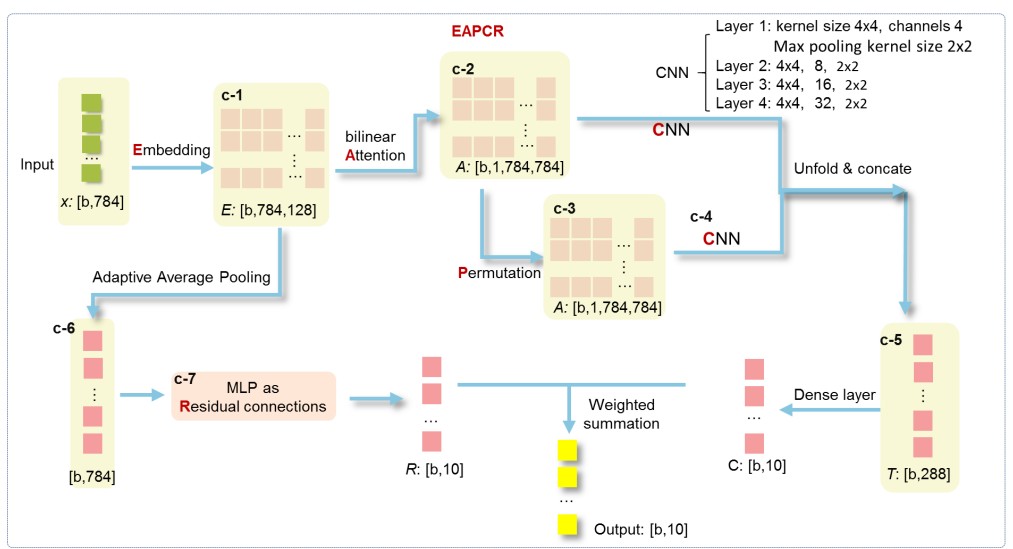

Figure 6: An illustration of the EAPCR, showing the detailed structure and parameter setting for the synthesized data without FRPs.

Table 5: The details of EAPCR and its ablation model EACR on synthesized data without FRPs.

| Model | Our ablation model EACR | | | Our model EAPCR | | |
|---|---|---|---|---|---|---|
| Parameters | 29151 | 59361 | 79236 | 37355 | 67565 | 87440 |
| Train | 30000 | 30000 | 30000 | 30000 | 30000 | 30000 |
| Test | 5000 | 5000 | 5000 | 5000 | 5000 | 5000 |
| Batch size | 64 | 64 | 64 | 64 | 64 | 64 |
| Epoch | 100 | 100 | 100 | 100 | 100 | 100 |
| Learning rate | 0.003 | 0.003 | 0.003 | 0.003 | 0.003 | 0.003 |
| Dropout | 0.5 | 0.5 | 0.5 | 0.5 | 0.5 | 0.5 |
| Embedding size | 128 | 128 | 128 | 128 | 128 | 128 |
| CNN Layer1 kernel size Channels Max pooling kernel size | Conv1(4x4, 4, 2x2) Conv2(4x4, 8, 2x2) Conv3(4x4, 16, 2x2) Conv4(4x4, 16, 2x2) | Conv1(4x4, 4, 2x2) Conv2(4x4, 8, 2x2) Conv3(4x4, 16, 2x2) Conv4(4x4, 16, 2x2) | Conv1(4x4, 4, 2x2) Conv2(4x4, 8, 2x2) Conv3(4x4, 16, 2x2) Conv4(4x4, 16, 2x2) | Conv1(4x4, 4, 2x2) Conv2(4x4, 8, 2x2) Conv3(4x4, 16, 2x2) Conv4(4x4, 16, 2x2) | Conv1(4x4, 4, 2x2) Conv2(4x4, 8, 2x2) Conv3(4x4, 16, 2x2) Conv4(4x4, 16, 2x2) | Conv1(4x4, 4, 2x2) Conv2(4x4, 8, 2x2) Conv3(4x4, 16, 2x2) Conv4(4x4, 16, 2x2) |
| CNN Layer2 kernel size Channels Max pooling kernel size | / | / | / | Conv1(4x4, 4, 2x2) Conv2(4x4, 8, 2x2) Conv3(4x4, 16, 2x2) Conv4(4x4, 16, 2x2) | Conv1( 4x4, 4, 2x2) Conv2(4x4, 8, 2x2) Conv3(4x4, 16, 2x2) Conv4(4x4, 16, 2x2) | Conv1(4x4, 4, 2x2) Conv2(4x4, 8, 2x2) Conv3(4x4, 16, 2x2) Conv4(4x4, 16, 2x2) |
| Residual | 784to26,26to10 | 784to64,64to10 | 784to89,89to10 | 784to26,26to10 | 784to64,64to10 | 784to89,89to10 |
| MLP | 144to10 | 144to10 | 144to10 | 288to10 | 288to10 | 288to10 |
| Permutation matrix size | / | / | / | 784×784 | 784×784 | 784×784 |

Table 6: The details of other models on synthesized data without FRPs.

| Model | MLP | CNN | GCN | Transformer |
|---|---|---|---|---|
| Train | 30000 | 30000 | 30000 | 30000 |
| Test | 5000 | 5000 | 5000 | 5000 |
| Batch size | 16 | 16 | 128 | 64 |
| Epoch | 100 | 100 | 1000 | 100 |
| Learning rate | 0.001 | 0.001 | 0.003 | 0.0003 |
| Dropout | 0.5 | 0.5 | 0.5 | 0.5 |
| Embedding | / | / | / | Embedding (2,128) |
| Feature extraction | / | Conv1(5x5,8,2x2) Conv2(5x5,16,2x2) | GCN1(1,64) GCN2(64,128) | Transformer (128,4,1) |
| Residual | / | / | Linear1(784,128) Linear2(128, 10) | Linear1(784, 64) Linear2(64, 10) |
| Classification layer | Linear1(784 ,64) Linear2(64,10) | Linear1(784, 64) Linear2(64, 10) | Linear1(128,10) | Linear1(784,128) Linear2(128,10) |

*GCNs* primarily extract features through the relationships between nodes within a graph structure. Unlike MLP's fully connected layers and CNN's convolutional layers, GCN use adjacency matrices and node feature matrices for feature extraction, leveraging the graph's local structure to capture relationships between nodes. Here we test GCN only on data without FRPs. When GCN with adjacency matrices (AM) given by the correlation matrix recovered from simple statistics on the data, its performance slightly improved, achieving an accuracy of 91.2%, as shown in Fig. 3-c-4. However, the classification accuracy of GCN with random AM drops to only 88.2%.

*RF* is an algorithm in machine learning that can consider the information gain from combinations of features. It enhances prediction accuracy and stability by constructing multiple DT and aggregating their predictions. Unlike MLP, RF can naturally account for the combinations and interactions between features when processing various types of data, thereby demonstrating certain effectiveness in feature extraction. Particularly in tasks that require capturing complex data structures and relationships, RF can utilize the structure of its DT to assess and exploit the information gain among features. In this experiment, as shown in Fig. 3-c-5, RF generally outperforms MLP. However, the advantage of RF over MLP is not obvious, showing only a slight improvement in performance.

*Our ablation model, EACR*, which differs from EAPCR with no permuted CNN, also achieved outstanding performance with fewer parameters than MLP, highlighting its advanced feature extraction capabilities. For example, experimental results demonstrate that despite using fewer parameters (for example, 29,151 parameters achieving 93.3% accuracy, 59,361 parameters achieving 94.3% accuracy, and 79,236 parameters achieving 95.2% accuracy, as shown in Fig. 3-c-6), EACR still exhibits exceptional performance in handling complex tasks, surpassing traditional models like MLP.

*KAN* is a neural network based on the Kolmogorov-Arnold representation theorem (Liu et al., 2024a). Compared with MLP, KAN does not only rely on fully connected layers to process data but builds a network formed by a combination of nested functions. This structure can more effectively capture and represent the complex features of the input data. However, our experimental results show that when the number of parameters is large, the effect of KAN does not reach the level of MLP. As shown in Fig. 3-c-10, when the number of parameters is about 100,000, the accuracy of KAN is less than 90%. However, when CNN is combined with KAN, the feature extraction capability of CNN can effectively extract local features and edge information from the image, and these features are used as input to KAN. The results show that this combination significantly improves the performance of the model. As shown in Fig. 3-c-8, when the number of parameters is 93,624, the accuracy is 97.5%. However, when processing data without FRPs, it is difficult for CNN to capture effective features for KAN to use, resulting in a decline in overall model performance. As shown in Fig. 3-c-9, under the same parameters, the accuracy rate is only 89.7%.

*Transformers* (Vaswani, 2017), through their attention mechanisms, are capable of capturing global dependencies among all elements in the input data. Transformers calculate the mutual influences of all pairs of elements within the input sequence using self-attention layers, providing high flexibility and strong capability for information integration. However, this mechanism also leads to a significant increase in the number of model parameters. Despite the large number of parameters,

Transformers did not significantly outperform MLP in our experiments on new types of image processing tasks. Specifically, even though a Transformer was configured with 1,252,949 parameters, its accuracy was the same as that of an MLP configured with 101,770 parameters, both achieving 94.4% (Fig. 3-c-11).

The specific parameters and accuracy of MLP, CNN, GCN, RF, DT, Transformers,Least absolute shrinkage and selection operator (LASSO), Elastic Net Regression (Elastic Net),our ablation method EACR and our method EAPCR results are shown in the Tab. 7.

Table 7: Experimental results of MLP, CNN,GCN,RF,DT, Transformers, our ablation method EACR, and our method EAPCR.

| Raw Image (data with FRPs) | | | Synthesized data (data without FRPs) | | |
|---|---|---|---|---|---|
| Method | Parameter quantity | Accuracy | Method | Parameter quantity | Accuracy |
| MLP-1 | 20680 | 90.5% | MLP-1 | 20680 | 90.6% |
| MLP-2 | 31810 | 91.4% | MLP-2 | 31810 | 91.5% |
| MLP-3 | 41350 | 92.3% | MLP-3 | 41350 | 92.0% |
| MLP-4 | 50890 | 92.7% | MLP-4 | 50890 | 92.8% |
| MLP-5 | 63610 | 93.5% | MLP-5 | 63610 | 93.5% |
| MLP-6 | 71560 | 93.7% | MLP-6 | 71560 | 93.6% |
| MLP-7 | 83485 | 94.2% | MLP-7 | 83485 | 93.9% |
| MLP-8 | 91435 | 94.2% | MLP-8 | 91435 | 94.3% |
| MLP-9 | 101770 | 94.3% | MLP-9 | 101770 | 94.4% |
| RF-1 | 28729 | 92.0% | RF-1 | 29241 | 91.7% |
| RF-2 | 57831 | 93.7% | RF-2 | 58390 | 93.6% |
| RF-3 | 87038 | 94.2% | RF-3 | 87377 | 94.2% |
| RF-4 | 116294 | 94.7% | RF-4 | 116168 | 94.2% |
| RF-5 | 145773 | 94.7% | RF-5 | 145613 | 94.8% |
| RF-6 | 291533 | 94.9% | RF-6 | 292517 | 94.9% |
| RF-7 | 584091 | 95.2% | RF-7 | 583980 | 95.2% |
| RF-8 | 876758 | 95.2% | RF-8 | 875610 | 95.1% |
| RF-9 | 1169481 | 95.3% | RF-9 | 1168306 | 95.1% |
| RF-10 | 1461923 | 95.4% | RF-10 | 1461973 | 95.1% |
| DT | 4815 | 83.3% | DT | 4815 | 83.1% |
| LASSO | 7850 | 88.7% | LASSO | 7850 | 88.8% |
| Elastic Net | 7850 | 88.9% | Elastic Net | 7850 | 88.9% |
| KAN | 91390 | 87.4% | KAN | 91390 | 88.0% |
| KAN | 118386 | 86.6% | KAN | 118386 | 87.5% |
| CNN+KAN | 93624 | 97.5% | CNN+KAN | 93624 | 89.7% |
| CNN+MLP | 21928 | 97.7% | CNN+MLP | 21928 | 90.8% |
| / | / | / | GCN with Random AM | 22451 | 88.2% |
| / | / | / | GCN with AM given by correlation matrix | 22451 | 91.2% |
| / | / | / | Transformer | 1252949 | 94.4% |
| / | / | / | **Ablation EACR-1** | **29151** | **93.3%** |
| / | / | / | **Ablation EACR-2** | **59361** | **94.3%** |
| / | / | / | **Ablation EACR-3** | **79236** | **95.2%** |
| / | / | / | **Our EAPCR-1** | **37355** | **94.5%** |
| / | / | / | **Our EAPCR-2** | **67565** | **95.0%** |
| / | / | / | **Our EAPCR-3** | **87440** | **95.5%** |

# E EXPERIMENT DETAILS: SCIENTIFIC TASKS

Tab. 8 provides details of the heart disease dataset from the UCI Machine Learning Repository, consisting of 13 feature attributes and 1 target attribute. The attributes are as follows: age (age in years), sex (1 = male, 0 = female), cp (chest pain type, where 1 = typical angina, 2 = atypical angina, 3 = non-anginal pain, 4 = asymptomatic), trestbps (resting blood pressure in mm Hg on admission to the hospital), chol (serum cholesterol in mg/dl), fbs (fasting blood sugar > 120 mg/dl, 1 = true, 0 = false), restecg (resting electrocardiographic results, where 0 = normal, 1 = having ST-T wave abnormality such as T wave inversions or ST elevation/depression > 0.05 mV, 2 = showing probable or definite left ventricular hypertrophy by Estes' criteria), thalach (maximum heart rate achieved), exang (exercise-induced angina, 1 = yes, 0 = no), oldpeak (ST depression induced by exercise relative to rest), slope (slope of the peak exercise ST segment, where 1 = upsloping, 2 = flat, 3 = downsloping), ca (number of major vessels colored by fluoroscopy, ranging from 0 to 3), thal (where 3 = normal, 6 = fixed defect, 7 = reversible defect). The target attribute indicates the presence of heart disease, where 0 represents no disease and 1 represents disease.

Tab. 9 provides details of the inorganic catalysis dataset. The value ranges for the characteristic variables are as follows: Dopant (Ag, Bi, C, Ce, Cd, F, Fe, Ga, I, Mo, N, Ni, S), Dopant/Ti mole ratio (0–93:5), Calcination temperature (400–900°C), Pollutant (Methylene blue [MB], phenol, rhodamine B [RhB], methyl orange [MO], methyl red [MR], acid orange [AO]), Catalyst/pollutant mass ratio (5:1–1000:1), pH (2–13), Experimental temperature (16–32°C), Light wavelength (254–600 nm), and Illumination time (5–480 minutes).

Tab. 10 provides details of the sensor dataset. The dataset includes the following variables: footfall (the number of people or objects passing by the machine), tempMode (the temperature mode or setting of the machine), AQ (air quality index near the machine), USS (ultrasonic sensor data, indicating proximity measurements), CS (current sensor readings, indicating the electrical current usage of the machine), VOC (volatile organic compounds level detected near the machine), RP (rotational position or RPM of the machine parts), IP (input pressure to the machine), Temperature (the operating temperature of the machine), and fail (binary indicator of machine failure, where 1 indicates failure and 0 indicates no failure).

Table 8: UCI heart disease dataset

| age | sex | cp | trestbps | chol | fbs | restecg | thalach | exang | oldpeak | slope | ca | thal | target |
|-----|-----|-----|----------|------|-----|---------|---------|-------|---------|-------|-----|------|--------|
| 63 | 1 | 1 | 145 | 233 | 1 | 2 | 150 | 0 | 2.3 | 3 | 0 | 6 | 0 |
| 67 | 1 | 4 | 160 | 286 | 0 | 2 | 108 | 1 | 1.5 | 2 | 3 | 3 | 2 |
| 67 | 1 | 4 | 120 | 229 | 0 | 2 | 129 | 1 | 2.6 | 2 | 2 | 7 | 1 |
| … | … | … | … | … | … | … | … | … | … | … | … | … | … |
| … | … | … | … | … | … | … | … | … | … | … | … | … | … |
| … | … | … | … | … | … | … | … | … | … | … | … | … | … |
| 37 | 1 | 3 | 130 | 250 | 0 | 0 | 187 | 0 | 3.5 | 3 | 0 | 3 | 0 |
| 41 | 0 | 2 | 130 | 204 | 0 | 2 | 172 | 0 | 1.4 | 1 | 0 | 3 | 0 |
| 56 | 1 | 2 | 120 | 236 | 0 | 0 | 178 | 0 | 0.8 | 1 | 0 | 3 | 0 |
| 62 | 0 | 4 | 140 | 268 | 0 | 2 | 160 | 0 | 3.6 | 3 | 2 | 3 | 3 |

Table 9: Inorganic catalysis dataset

| Dopant | Dopant/Ti mole ratio | Calcination temperature | Pollutant | Catalyst/Pollutant mass ratio | pH | Experimental temperature | Light wavelength | Illumination time | Degradation rate |
|--------|----------------------|-------------------------|-----------|-------------------------------|-----|--------------------------|------------------|-------------------|------------------|
| C | 16 | 400 | MB | 100 | 7 | 25 | 425 | 20 | 30 |
| C | 16 | 400 | MB | 100 | 7 | 25 | 425 | 40 | 43 |
| C | 16 | 400 | MB | 100 | 7 | 25 | 425 | 60 | 48 |
| … | … | … | … | … | … | … | … | … | … |
| … | … | … | … | … | … | … | … | … | … |
| … | … | … | … | … | … | … | … | … | … |
| Fe | 0 | 500 | MB | 100 | 7 | 0 | 545 | 60 | 28.1 |
| Fe | 0 | 500 | MB | 100 | 7 | 0 | 545 | 120 | 44 |
| Fe | 0 | 500 | MB | 100 | 7 | 0 | 545 | 180 | 52.8 |
| Fe | 0 | 500 | MB | 100 | 7 | 0 | 545 | 300 | 69 |

Table 10: Sensor measurements dataset

| footfall | tempMode | AQ | USS | CS | VOC | RP | IP | Temperature | fail |
|----------|----------|-----|-----|-----|-----|-----|-----|-------------|------|
| 0 | 7 | 7 | 1 | 6 | 6 | 36 | 3 | 1 | 1 |
| 190 | 1 | 3 | 3 | 5 | 1 | 20 | 4 | 1 | 0 |
| 5 | 5 | 3 | 3 | 6 | 1 | 24 | 6 | 1 | 0 |
| ... | ... | ... | ... | ... | ... | ... | ... | ... | ... |
| ... | ... | ... | ... | ... | ... | ... | ... | ... | ... |
| ... | ... | ... | ... | ... | ... | ... | ... | ... | ... |
| 74 | 7 | 4 | 4 | 7 | 2 | 88 | 2 | 2 | 0 |
| 190 | 0 | 2 | 4 | 6 | 2 | 20 | 4 | 2 | 0 |
| 12 | 3 | 4 | 6 | 3 | 2 | 27 | 3 | 2 | 0 |
| 0 | 7 | 6 | 1 | 6 | 6 | 44 | 4 | 2 | 1 |

We also applied our EAPCR method to additional non-image medical data, inorganic catalysis data, and system anomaly detection data, with the specific results shown below.

**More non-image medical diagnosis:** Data1: Lung cancer dataset (Mamun et al., 2022), which contains 309 samples, 15 feature attributes, and 1 classification attribute. The feature attributes include: Gender (M = male, F = female), Smoking (YES = 2, NO = 1), Yellow fingers (YES = 2, NO = 1), Anxiety (YES = 2, NO = 1), Peer pressure (YES = 2, NO = 1), Chronic disease (YES = 2, NO = 1), Fatigue (YES = 2, NO = 1), Allergy (YES = 2, NO = 1), Wheezing (YES = 2, NO = 1), Alcohol consumption (YES = 2, NO = 1), Coughing (YES = 2, NO = 1), Shortness of breath (YES = 2, NO = 1), Swallowing difficulty (YES = 2, NO = 1), and Chest pain (YES = 2, NO = 1). The classification attribute is Lung Cancer (YES, NO). The specific results are shown in Tab. 11.

Data2: Breast cancer dataset (https://www.kaggle.com/datasets/abdelrahman16/breast-cancer-prediction), which contains 213 samples and 9 feature attributes. The attributes include: Year (the year when the data was recorded), Age (age of the patient), Menopause (menopausal status of the patient, 1 for postmenopausal, 0 for premenopausal), Tumor Size (size of the tumor in centimeters), Inv-Nodes (presence of invasive lymph nodes), Breast (breast affected: Left or Right), Metastasis (presence of metastasis, 0 for no, 1 for yes), Breast Quadrant (quadrant of the breast where the tumor is located, e.g., Upper inner, Upper outer), History (patient's history of breast cancer, 0 for no, 1 for yes), and Diagnosis Result (Benign or Malignant). The specific experimental results are shown in Tab. 12.

Table 11: Comparison of our method with others in the diagnosis of Lung cancer dataset

| Method | Accuracy | Precision | Recall | F1 Score | AUC |
|--------|----------|-----------|--------|----------|-----|
| Bagging | 89.76% | 91.88% | 89.35% | 90.00% | 95.30% |
| AdaBoost | 90.70% | 90.70% | 90.70% | 90.70% | 97.62% |
| LightGBM | 92.56% | 93.93% | 92.10% | 92.71% | 92.71% |
| XGBoost | 94.42% | 95.66% | 94.46% | 94.74% | 98.14% |
| **EAPCR** | **96.30%** | **96.49%** | **96.49%** | **96.49%** | **98.61%** |

Table 12: Comparison of our method with others in the diagnosis of Breast cancer dataset

| Method | Accuracy | Precision | Recall | F1 Score |
|--------|----------|-----------|--------|----------|
| XGBoost | 83.72% | 81.25% | 76.47% | 78.79% |
| DT | 83.72% | 81.25% | 76.47% | 78.79% |
| KNN | 86.05% | 82.35% | 82.35% | 82.35% |
| Logistic Regression | 88.37% | 87.50% | 82.35% | 84.85% |
| Extra Trees Classifier | 88.37% | 87.50% | 82.35% | 84.85% |
| **EAPCR** | **93.02%** | **100%** | **82.35%** | **90.32%** |

**More inorganic catalysis efficiency prediction:** Data 1: Thermocatalytic dataset (Schossler et al., 2023), which includes three metal elements (M1, M2, M3), support ID, M1M2M3 ratio, temperature, volume flow rate, methane flow rate, time, and methane/O2 ratio. The specific experimental results are shown in Tab. 13.

Data 2: Each sample in Dataset 2 (Puliyanda, 2024) contains 8 experimental variables, including organic pollutants (OC), UV light intensity (I, mW/cm$^2$), wavelength (W, nm), dosage (D, mg/cm$^2$), humidity (H, %), experimental temperature (T, °C), reactor volume (R, L), and initial concentration of pollutants (InitialC, ppmv). The light intensity ranges from 0.36 to 75 mW/cm$^2$, the illumination wavelength ranges from 253.7 to 370 nm, the titanium dioxide dosage ranges from 0.012 to 5.427 mg/cm$^2$, the humidity ranges from 0 to 1600%, the experimental temperature ranges from 22 to 350°C, the reactor volume ranges from 0.04 to 216 L, and the initial concentration of pollutants ranges from 0.001 to 5944 ppmv. The photodegradation rate (k, min$^{-1}$/cm$^2$) is set as the response variable. The specific experimental results are shown in Tab. 14.

Table 13: Comparison of our method with others in the Inorganic catalysis data set.

| Model | MAE | MSE | RMSE | $R^2$ |
|-------|-----|-----|------|-------|
| ANN with BO (Puliyanda, 2024) | / | 0.559 | / | 0.438 |
| Catboost + adaboost (Puliyanda, 2024) | / | 0.064 | / | 0.922 |
| GBM with BO (Puliyanda, 2024) | / | 0.117 | / | 0.882 |
| XGB with HYPEROPT (Puliyanda, 2024) | / | 0.073 | / | 0.927 |
| **EAPCR** | **0.131 ± 0.004** | **0.054 ± 0.001** | **0.233 ± 0.003** | **0.940 ± 0.002** |

Table 14: Comparison of our method with others in the Inorganic catalysis data set.

| Model | MAE | MSE | RMSE | $R^2$ |
|-------|-----|-----|------|-------|
| RF | / | / | 3.40 | 0.89 |
| **EAPCR** | **1.27 ± 0.02** | **2.88 ± 0.09** | **1.69 ± 0.02** | **0.97 ± 0.00** |

**More system anomaly detection:** Centrifugal pumps dataset (Mallioris et al., 2024). The dataset contains 5,118 rows of measurements from two centrifugal pumps from the same manufacturer. These measurements include key features: value_ISO, value_DEM, value_ACC, value_P2P, value_TEMP, minute, second, year, month, day, hour, and Machine_ID (1 for healthy, 2 for maintenance status). The specific results are shown in Tab. 15.

Table 15: Comparison of our method with others in the diagnosis of Centrifugal pumps dataset.

| Method | Accuracy | Precision | Recall | F1 Score |
|--------|----------|-----------|--------|----------|
| SVM | 96.51% | 97.77% | 95.50% | 96.66% |
| NB | 96.51% | 97.77% | 95.50% | 96.66% |
| RF | 98.25% | 96.73% | 100% | 98.33% |
| XGBoost | 98.83% | 97.82% | 100% | 98.89% |
| **EAPCR** | **100%** | **100%** | **100%** | **100%** |

# F  WHY FRPS MATTERS?

In this study, we show that FRPs are an important type of prior knowledge for the application of deep learning methods in various tasks. Another well-recognized prior knowledge is the annotated dataset. Despite significant research on overcoming the scarcity of annotated data in various scientific applications (Zhou et al., 2022; Gao et al., 2023; Liu et al., 2023), fewer efforts have focused on designing feature extractors without prior knowledge of FRPs.

The key to designing an effective feature extractor is to effectively sample the feature combinations consisting of strongly interactive features where the combined effect exceeds the sum of their individual contributions (Koh & Liang, 2017; Ali et al., 2012; Beraha et al., 2019; Deng et al., 2022). In this section, we demonstrate that these interactive feature combinations are selected from the combinations of correlated features. Typically, FRPs contain the underlying correlation information between features. As a result, combinations of related features often lead to combinations of correlated features, highlighting the importance of FRPs.

Here, we prove that these interactive features are correlated.

**Proposition F.1.** *If features $A$ and $B$ are independent, then:*

$$IG(Y, A) + IG(Y, B) = IG(Y, A, B)$$

*where:*

$$IG(Y, A) \triangleq H(Y) - H(Y|A)$$

$$IG(Y, B) \triangleq H(Y) - H(Y|B)$$

$$IG(Y, A, B) \triangleq H(Y) - H(Y|A, B)$$

*with $H(Y)$ the entropy of $Y$, $IG(Y, A)$ the information gain of $Y$ given $A$, $H(Y|A)$ the conditional entropy of $Y$ given $A$, and $IG(Y, A, B)$ the information gain of $Y$ given $A$ and $B$*

*Proof.* Since $A$ and $B$ are independent, their information contribution to $Y$ is completely independent; therefore, their joint effect equals the simple sum of their individual effects minus the entropy of $Y$, i.e.:

$$H(Y|A, B) = H(Y|A) + H(Y|B) - H(Y)$$

Substituting the definition of information gain into the assumption of independent conditional entropy:

$$IG(Y, A, B) = H(Y) - H(Y|A, B) = H(Y) - (H(Y|A) + H(Y|B) - H(Y))$$

$$IG(Y, A, B) = 2H(Y) - H(Y|A) - H(Y|B)$$

The sum of information gains:

$$IG(Y, A) + IG(Y, B) = (H(Y) - H(Y|A)) + (H(Y) - H(Y|B)) = 2H(Y) - H(Y|A) - H(Y|B)$$

Thus, it is proved that $IG(Y, A) + IG(Y, B) = IG(Y, A, B)$. $\qquad\square$

**Proposition F.2.** *If:*

$$IG(Y, A, B) > IG(Y, A) + IG(Y, B), \tag{3}$$

*then, features $A$ and $B$ are correlated.*

*Proof.* Rearranging equation 3 gives us

$$IG(Y, A, B) > IG(Y, A) + IG(Y, B)$$

Following the definition of information gain, we have:

$$H(Y) - H(Y|A, B) > (H(Y) - H(Y|A)) + (H(Y) - H(Y|B))$$

$$H(Y) - H(Y|A, B) > 2H(Y) - H(Y|A) - H(Y|B)$$

$$H(Y|A) + H(Y|B) - H(Y) > H(Y|A, B)$$

The above inequalities imply that $A$ and $B$ jointly provide more information than the sum of the information provided individually. This is typically because there is some interaction or dependency between $A$ and $B$ that causes their combined information gain to exceed the individual gains, hence $A$ and $B$ are not independent. $\qquad\square$

**Proposition F.3.** *If feature $A$ and $B$ have an interaction, i.e.:*

$$H(Y|A, B) < H(Y|A) + H(Y|B) - H(Y), \tag{4}$$

*then:*

$$IG(Y, A, B) > IG(Y, A) + IG(Y, B)$$

*Proof.* Rewriting information gain:

$$IG(Y, A) + IG(Y, B) = (H(Y) - H(Y|A)) + (H(Y) - H(Y|B)) = 2H(Y) - H(Y|A) - H(Y|B)$$

Substituting equation 4 into the calculation of information gain gives:

$$IG(Y, A, B) = H(Y) - H(Y|A, B)$$

$$IG(Y, A, B) > H(Y) - (H(Y|A) + H(Y|B) - H(Y))$$

$$IG(Y, A, B) > 2H(Y) - H(Y|A) - H(Y|B)$$

This implies that the joint information gain $IG(Y, A, B)$ exceeds the sum of the individual gains. This indicates a positive interaction between features $A$ and $B$ in influencing $Y$, where their combined impact reduces the uncertainty of $Y$ more than their individual effects summed simply. $\qquad\square$

**Design a feature extractor with FRPs.** If the feature correlation patterns are embedded in the FRPs, it is only necessary to sample the combinations of features that are known to be correlated, Because those features that are not correlated do not have interaction effects, there is no need to consider their combinations, as proved in F.2. The sampling scope will be largely limited.

**The challenge of designing a feature extractor without FRPs.** When the FRPs is unknown correlation patterns among features are unknown, for $N$ features, combinations of different features need to be considered. For example, when sampling one feature, the number of samplings is $C_N^1$, when sampling two features, it is $C_N^2$, and when sampling all $N$ features, it is $C_N^N$. Therefore, for a sample composed of $N$ features, the total number of samplings required is: $C_N^1 + C_N^2 + C_N^3 + \cdots + C_N^N = 2^N - 1$.

## G EAPCR RECOVERS THE HIDDEN FRPS

The effectiveness of our method lies in its ability to accurately reconstruct the correlation patterns within the data, even when these patterns become less apparent or lost after transformations. As shown in Fig. 7, whether the images are in their original state or transformed, the matrices reconstructed by our EAPCR model consistently reflect the true pixel correlation patterns with a recall rate of $84.6\%$. This highlights the model's precision and reliability in restoring these patterns, even when pixel positions are altered.

In a more detailed analysis, we observed a $55.0\%$ recall rate when comparing the correlation matrix of transformed images to the original pixel patterns. Conversely, comparing the correlation matrix from the original images to the transformed image patterns yielded a recall rate of $66.3\%$. This indicates that for data containing different feature correlation patterns, the patterns restored by our model vary significantly. **This demonstrates that our EAPCR is capable of adaptively restoring the unique hidden relationships between features.** EAPCR shows a strong ability to recover underlying data relationships, further emphasizing its robustness and effectiveness in scenarios where explicit correlation patterns are not directly accessible.

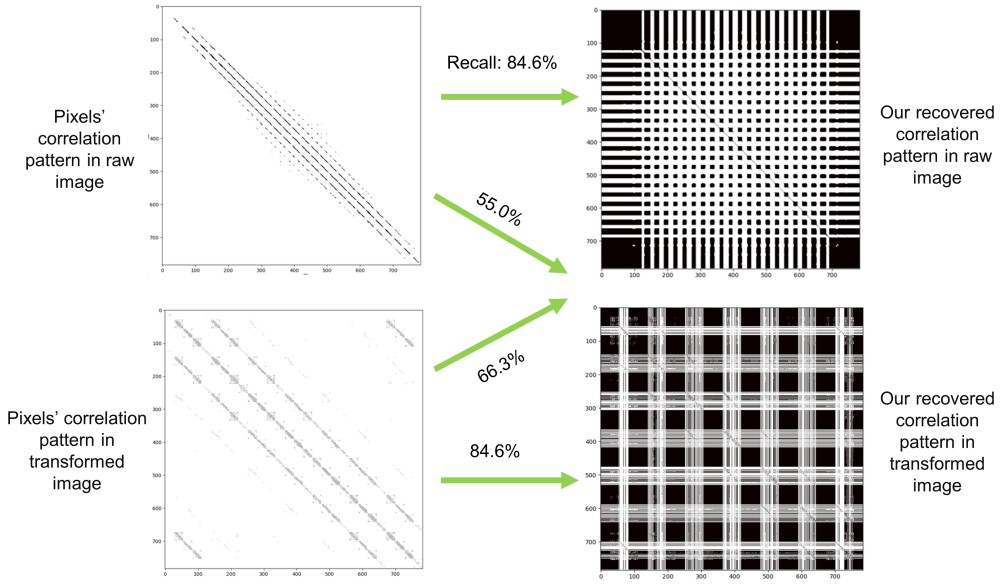

Figure 7: The alignment between the pixel correlation matrices reconstructed using our technique and the actual pixel correlation patterns in both original and transformed images is demonstrated. The correlation matrices were binarized.

## H  DATA AND CODE

The public datasets can be found in the corresponding references. Other data and code will be released.

The experiments were conducted on devices with the following specifications. GPU: `NVIDIA GeForce RTX 3090` with `24GB`, `RTX4080` with `16GB`, and `RTX4090` with `24GB` of VRAM, CPU: `13th Gen Intel(R) Core(TM) i9-13900K`. The source code is freely available at the `GitHub` repository after the peer review process.

