# OpenReview forum: "EAPCR: A Universal Feature Extractor for Scientific Data Without Explicit Feature Relation Patterns"
_ICLR.cc/2025/Conference — ICLR 2025 Conference Withdrawn Submission_

### Official Review · Reviewer_NWah · 2024-10-28

**Soundness:** 2
**Presentation:** 2
**Contribution:** 2
**Rating:** 3
**Confidence:** 2

**Summary:**

In this work, the authors introduce EAPCR, a universal feature extractor designed for data lacking explicit relationships between features. This absence of explicit relationships often poses challenges for deep learning models on tabular datasets, where traditional ML models like decision trees tend to excel. The authors demonstrate the effectiveness and robustness of EAPCR on UCI datasets as well as on synthetic datasets.

**Strengths:**

- The problem definition is very important and is relevant to the ML community and practitioners.
- The method seems general and can be used in various settings.
- Empirical results look strong.

**Weaknesses:**

- Since the permutation $\mathbf{M}$ plays such a key role, please provide some details or intuition in the main paper on how to create these permutations for various datasets and modalities.
- Regarding the comment on GCN performance deteriorating on random adjacency matrices (line 428), this is not surprising. However to show that EAPCR is robust to these changes, the authors should use benchmark graph datasets and validate their findings.
- For tabular datasets, LightGBM or XGB would be better baselines (table 1).
- This method seems extremely computationally expensive. For a $N \times N$ image, it creates a permutation matrix of shape $N^2 \times N^2$.
- For tabular datasets, it is not clear how the binning is performed as creating a very fine/coarse binning may affect the values of the design matrix.

**Questions:**

In addition to the questions above, I have a few more questions :

-  How do you handle missing values and outliers?
- I am still confused by why you need to convert all features into categorical features? For ex. how do you handle the iris dataset?
- Related to the question above : how do you handle a noisy pinwheel dataset with different colors and the objective is to predict the color based on the coordinates?
- What kind of relations does our method learn in various tabular datasets?
- Without error bars, I am not sure if results in Table 3 are statistically significant. Also Table 3 is mostly empty.
- I am quite confused by Fig 3. An MLP on MNIST gets around 98%. The numbers reported are quite low.
- I may be missing something but what is the utility of graph datasets where the adjacency matrices are random and image datasets where the pixels are randomly shuffled.
- How is your method related to SSL? For example, treat $\mathbf{A}$ and permuted $\mathbf{A}$ as positive pairs.
- Related to the works on vision datasets, how does your method relate to MLP mixer [1]?

[1] MLP-Mixer: An all-MLP Architecture for Vision. Tolstikhin et al. NeurIPS 2021

---

### Official Review · Reviewer_p9or · 2024-10-30

**Soundness:** 2
**Presentation:** 3
**Contribution:** 2
**Rating:** 3
**Confidence:** 4

**Summary:**

The paper introduces EAPCR, a feature extraction method aimed at handling scientific datasets that lack explicit Feature Relation Patterns. EAPCR employs a permutation-based approach to transform feature relationships, followed by CNNs to capture both local and non-local correlations. The model is evaluated across multiple tasks and synthetic datasets, where it reportedly outperforms traditional ML and DL methods.

**Strengths:**

●	The paper targets the problem of feature extraction in scientific data where explicit relational structures are absent, a common issue in many applications.
●	EAPCR is extensively tested across various datasets, and the results consistently show its advantages over several traditional baselines.

**Weaknesses:**

●	The paper lacks comparisons with simple DL models on the UCI Cleveland heart disease and photocatalyst datasets, such as a small MLP without inductive biases. The ML baselines are based on rigid assumptions, while EAPCR leverages a shallow MLP to generate the embeddings. The current comparisons make it difficult to determine the true source of the performance gains.
●	The need for permutation before applying the CNN is not well justified. A multi-layer CNN should, in theory, learn both local and global correlations without this step. There is no strong theoretical argument provided for why permutation should improve the ability to capture feature interactions over standard CNN techniques.
●	More importantly, if a single permutation is helpful, why not use multiple permutations concatenated before feeding them into CNNs?
●	Another concern is that shuffling or permuting data to generate synthetic datasets disrupts important structural information, which does not reflect real-world challenges where meaningful relationships are usually present.
●	The details about how the permutation matrix is constructed should be moved to the main body of the paper, given its central role in the proposed method.
●	It is uncertain whether the superiority of EAPCR would persist as the dataset size and model complexity increase.

**Questions:**

none

---

### Official Review · Reviewer_7q7N · 2024-11-03

**Soundness:** 2
**Presentation:** 2
**Contribution:** 2
**Rating:** 5
**Confidence:** 3

**Summary:**

The paper introduces EAPCR, a universal feature extractor designed for scientific data that lacks explicit Feature Relation Patterns (FRPs), which are spatial, sequential, or relational correlations typically leveraged by deep learning models. Traditional machine learning methods, like Decision Trees and their variants, have been more successful than deep learning models in such scientific tasks due to this absence of clear feature relationships.

**Strengths:**

Innovative approach to feature extraction: EAPCR’s ability to handle data without explicit Feature Relation Patterns (FRPs) is a significant advantage, enabling deep learning to be applied effectively in scientific domains traditionally dominated by simpler machine learning models.

Improved generalization and adaptability: By exposing and sampling all potential feature interactions, EAPCR offers a universal solution applicable to a wide range of scientific tasks, making it valuable for fields like healthcare and anomaly detection where deep learning previously struggled.

**Weaknesses:**

1. What is the difference between EAPCR and creating a all connected graphs with each feature as a node? There are lot of work that creating graph with unknown edge relations, for example, use correlation and etc. Given this, I think this work is limited on innovation.

2. If you do want to claim that EAPCR is a better model compared to many other algorithms, it is more fair to compare EAPCR with graph algorithms like different GNNs for experiments in Table1 and 2.

3. Some of the experiments are using Kaggle datasets. I know that lot of kaggle datasets can improve performance by simply ensemble different simple ML algorithms. So this should also be included in experiments as a baseline.

4. Will this cost too much for high dimensional features? It will be better to include some complexity analysis here.

**Questions:**

See the weakness above

---

### Official Review · Reviewer_h2i4 · 2024-11-03

**Soundness:** 2
**Presentation:** 3
**Contribution:** 2
**Rating:** 5
**Confidence:** 3

**Summary:**

The paper proposes a method to learn the representation of structured scientific data with no explicit connection among features. The method employs bilinear attention and permuted CNN to understand the implicit structure. The method achieves better performance on three real-world datasets and one synthetic dataset.

**Strengths:**

- The paper addresses an important machine learning challenge within the domain of structured scientific data, and the motivation for tackling this problem is clearly articulated.

- The proposed method and presentation that makes both the approach and the experiments easy to follow.

**Weaknesses:**

- The paper defines the term "feature relationship patterns" but fails to explain the concept well. It is unclear what relationship the proposed method learns among features. Also, the paper only experiments with the proposed method on three datasets, which is not sufficient to call a method "universal".

- The experiments are not comprehensive. In Table 1, there's no comparison with RF and XGBosst. In Table 3, there is no XGBoost. Also, no NN-based methods are shown in Tables 1, 2, 3 (e.g. MLP, Transformer). How the conclusion comes "While Kolmogorov–Arnold Network (KAN) and feature extractors like Convolutional Neural Networks (CNNs), Graph Convolutional Networks (GCNs), and Transformers struggle".

- The synthetic experiment does not sound reasonable. Permuting patches in an image does corrupt the spatial relationship within an image, but it is irrelevant to the scientific data. The permutation corruption may highly overfit to the design of permuted CNN in the proposed method.

- The proposed method is quite similar to a Transformer, but the paper fails to explain why it performs better than the Transformer. The only comparison is in Figure 3, but it is unclear which positional encoder, masks, number of layers and heads are used here. A Transformer can always be designed to learn different relationships. It is unclear that the benchmark in Figure 3 is the best a Transformer can do.

**Questions:**

See weakness. The author needs to justify the reason for experimental design and why they can support the augments made.

---

### Note · Authors · 2024-11-12

**Comment:**

Thanks for the review. We decide to withdraw.

**Withdrawal Confirmation:**

I have read and agree with the venue's withdrawal policy on behalf of myself and my co-authors.